# Characterization of the Dynamic Properties of Clay–Gravel Mixtures at Low Strain Level

**Xianwen Huang [1], Aizhao Zhou [1,\*], Wei Wang [2,\*]**  **and Pengming Jiang [1]**

1   Department of Civil and Architecture Engineering, Jiangsu University of Science and Technology, Zhenjiang 212003, China; huangxianwen194@163.com (X.H.); jkdkjcjpm@163.com (P.J.)
2   School of Civil Engineering, Shaoxing University, Shaoxing 312000, China
*   Correspondence: zhouaizhao@126.com (A.Z.); wellswang@usx.edu.cn (W.W.)

**Abstract:** In order to support the dynamic design of subgrade filling engineering, an experiment on the dynamic shear modulus (G) and damping ratio (D) of clay–gravel mixtures (CGMs) was carried out. Forty-two groups of resonant column tests were conducted to explore the effects of gravel content (0%, 10%, 20%, 30%, 40%, 50%, and 60%, which was the mass ratio of gravel to clay), gravel shape (round and angular gravels), and confining pressure (100, 200, and 300 kPa) on the dynamic shear modulus, and damping ratio of CGMs under the same compacting power. The test results showed that, with the increase of gravel content, the maximum dynamic shear modulus of CGMs increases, the referent shear strain increases linearly, and the minimum and maximum damping ratios decrease gradually. In CGMs with round gravels, the maximum dynamic shear modulus and the maximum damping ratio are greater, and the referent shear strain and the minimum damping ratio are smaller, compared to those with angular gravels. With the increase of confining pressure, the maximum dynamic shear modulus and the referent shear strain increase nonlinearly, while the minimum and maximum damping ratios decrease nonlinearly. The predicting equation for the dynamic shear modulus and the damping ratio of CGMs when considering confining pressure, gravel content, and shape was established. The results of this research may put forward a solid foundation for engineering design considering low-strain-level mechanical performance.

**Keywords:** clay–gravel mixtures; dynamic shear modulus; damping ratio; resonant column; predicting equation

## 1. Introduction

Clay–gravel mixtures (CGMs) constitute a great engineering material, which is widely utilized in subgrade engineering, rock-fill dams, and sea filling [1–4]. A CGM is a heterogeneous two-phase material, consisting of low-strength clay and high-strength and large-sized gravels. As shown in Figure 1, subgrade engineering in Ganzhou, Jiangxi province, China, uses fillers composed of CGMs, in which the clay and gravel are sourced locally and from the nearby mountains, respectively. In construction engineering, because the excavated clay cannot meet the requirements of compactness, it is not usually used as backfill material, but transported to a special site for stacking as muck, which occupies precious land resources. CGMs can not only solve the problem of clay residue treatment, but can also meet the requirements of the compactness and mechanical properties of backfill materials, turn waste into engineering materials, and realize the sustainable utilization of construction resources. For meeting the traffic load requirements, designers not only need to consider the static properties of CGMs, but also consider the dynamic properties. A design based on dynamic stability is more in line with the real working conditions of subgrade materials. Compared with static design, it will extend the service life of subgrade materials, reduce the repair times, save building resources, and promote

the sustainable use of building resources. The dynamic shear modulus and the damping ratio are the two typical parameters in CGMs, the former reflecting the bearing capacity of materials under shear force and the latter expressing the amplitude attenuation of dynamic load in soil [5–8]. Hence, it is necessary to obtain the dynamic shear modulus and the damping ratio of CGMs for supporting engineering construction.

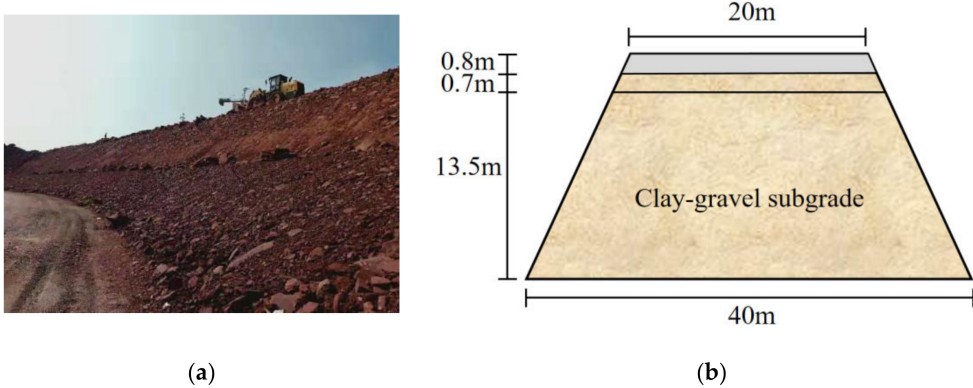

(a)                                                                                    (b)

**Figure 1.** Subgrade filling construction. (**a**) Subgrade filling construction site;(**b**) Geometric size of subgrade section.

Numerous experimental studies of the dynamic shear modulus and the damping ratio of soil have been conducted by researchers. Abdulkadir et al. [9] measured the dynamic shear modulus and the damping ratio of sand–mica mixtures, considering different mixture ratios of sand and mica. Okur et al. [10] studied the dynamic shear modulus of natural fine-grained soils under cyclic loading. Wang et al. [11] investigated the dynamic shear modulus and the damping ratio of sand soil considering fine contents. Chen et al. [12] conducted cyclic triaxial compression tests to investigate the dynamic properties of organic soil. Then, Zhang et al. [5,13] studied the dynamic properties of frozen silty soil with coarse-grained contents, and concluded that the increment of dynamic properties of frozen silty soil is positively correlated with coarse-grained content, which is similar to the conclusion of sand–gravel mixtures obtained by Xu et al. [14]. However, few studies have focused on CGMs, whose dynamic shear modulus and damping ratio have not been studied completely.

As a two-phase material, the impact of varying mixed ratios on dynamic properties has been studied by researchers. Wichtmann et al. [15] concluded that the maximum dynamic shear modulus decreased with increasing fine content in quartz sand. Xu et al. [11,14] also found that the fine particle content affected the dynamic properties, and corresponding relationship effects of gravel content to dynamic properties were put forward in sand–gravel mixtures. Except gravel content, Sun et al. [16] pointed out that the influence of granule shape on dynamic properties cannot be ignored, the importance of which has been verified by experiments using gravels. Medani et al. [17] studied the dynamic shear modulus and damping ratio of CGMs under the same mass density, and some conclusions regarding the effect of granular shape on dynamic properties of CGMs were obtained. Bouwman et al. [18–20] described the shape evaluation method of particles, and conducted some indoor dynamic experiments to verify the importance of granular shape for the dynamic properties of materials. In subgrade filling engineering with CGMs (which is a typical two-phase material), the gravel shape is not usually similar, and the difference in gravel shape leads to different dynamic properties, especially in a high-gravel content mixture. Hence, it is necessary to study the influence of granular shape on dynamic properties in CGMs for supporting engineering design.

Many monitoring results show that the dynamic strain of subgrade soil is smaller than $10^{-4}$ [21–24], at which range the resonant column device is advised to measure the dynamic properties of materials [25–27]. In this article, the traditional resonant column testing device was improved so that it can be used to test the samples without self-stability. Confining pressure, gravel content, and

shape were taken into account to design a CGM test and establish the predicting equation of dynamic shear modulus and damping ratio of CGMs. An electron microscope and Computer tomography (CT) were introduced to interpret the dynamic properties' mechanism of CGMs.

## 2. Preparation for Specimen and Test Scheme

### 2.1. Experimental Clay

The clay was taken from a subgrade construction site in Ganzhou, Jiangxi province, China, whose buried depth was 7 m. The samples experienced sun drying, breaking, selection, mixture with water, and sealed preservation, and the specific steps are shown in the following. According to the Chinese National Geotechnical Test Standard, GB/T 50123-1999, [28] the physical parameters and gradation curves of clay were obtained, which are shown in Table 1 and Figure 2, respectively.

**Table 1.** The physical parameters of clay.

| Item | Specific Gravity | Liquid Limit | Plastic Limit | Plasticity Index | Optimized Water Content | Expansion Ratio |
|---|---|---|---|---|---|---|
| Value | 2.75 | 43.2% | 20.2% | 23% | 15% | 41% |
| Standard deviation | 0.006 | 0.8% | 0.5% | - | 0.2% | 0.6% |

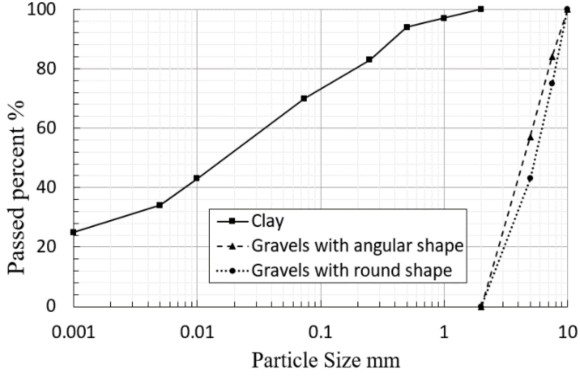

**Figure 2.** Gradation curves of clay and gravel.

Step 1 is sun drying: the obtained clay was broken into smaller-sized pieces and placed in the sun to reduce the water in the specimen.

Step 2 is breaking: the dried clay was broken into grains of sizes smaller than 2 mm. It is advised to use a jaw-type crusher or a rubber hammer in order to protect the original structural characteristics of the clay grains, instead of using crushers that break the clay blocks by cutting forces.

Step 3 is a selection of samples: select the clay grains smaller than 2 mm by particle screening.

Step 4 is a mixture with water: add water into the dried clay according to the optimum water content of clay for achieving maximum density.

Step 5 is storing: place the clay in black packaging and keep it in a cool place.

### 2.2. Experimental Gravels

In the design of subgrade filling engineering, the size of the gravels in CGMs is defined in the range of 2 to 40 mm, whereas the minimum size for testing a specimen in a resonant column device is 50 mm. If the CGM at the site is directly used to test the dynamic characteristics by a resonant column device, the size effect from large-sized gravels cannot be ignored, so it is necessary to conduct a reduced scale test. Based on the technical manuals of the geotechnical test and the corresponding references [2,29–31], it is concluded that the diameter of the gravels should not be greater than one

fifth of the minimum size of the specimen. Hence, in in-door model testing, the gravel size was limited within the range of 2 mm (which is the threshold between soil and gravel) and 10 mm (which is one fifth of the diameter of the specimen), and the specific gradation curves are shown in Figure 2.

In subgrade filling engineering, most sections of the road are filled with CGMa, which consists of clay and angular gravels, and others are filled with CGMr, consisting of clay and round gravels. Many studies have proved that the particle shape plays a significant role in the dynamic properties of materials. Hence, based on the classification of particle shape by Meidani et al. [17–19,32] and the in-site gravels' shape, two typical gravels were selected, including both round and angular shapes, as shown in Figure 3a,c.

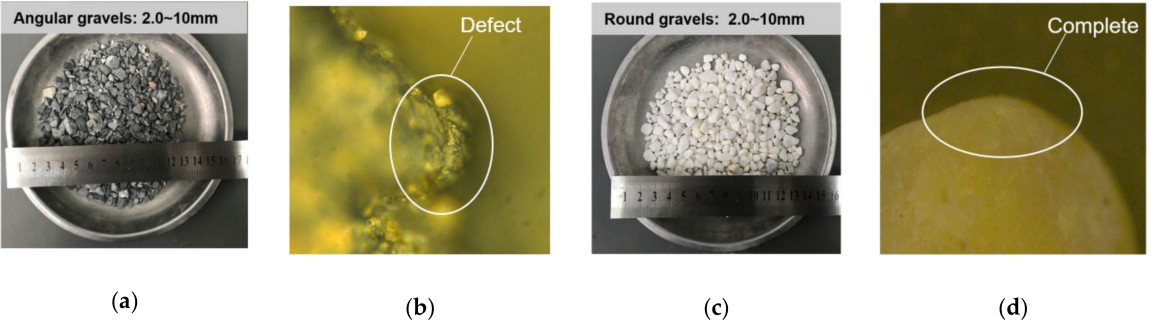

(**a**) (**b**) (**c**) (**d**)

**Figure 3.** Two typical types of gravel.(**a**) Angular gravels; (**b**) Magnified view of angular gravels; (**c**) Round gravels; (**d**) Magnified view of round gravels.

In this article, the angular gravels were screened from the rocks broken by a jaw crusher, and the round gravels were collected from rivers. As shown in Figure 3b,d, it can be found that the angular gravels have sharp edges and some defects near the edges; in contrast, the round gravels did not have sharp edges and had better completion near the edges. The insufficient breaking of rocks and grinding of small particles or water in rivers can be used to explain the differences in edge characteristics of angular and round gravels, respectively.

### 2.3. Preparation for Specimen

Compactness degree, density, and compacting power usually act as the control parameters in specimen preparation [14,25,29,33–35]. In this article, for better comparing the indoor tests with in-site engineering construction, the same compacting power (160 J, mass of compact hammer is 2 kg, g is recognized as 10 N/kg, height of compact hammer is 0.4 m, 20 times with the compacting-sample device of each layer) was used in specimen preparation. When preparing the specimen, the water content of the clay was controlled at 15% (based on the optimized water content of clay), and then the clay was mixed with wet gravels. After this, the mixtures were poured into cast iron modulus cylinders that were 50 × 100 mm (diameter × height). Within these cylinders, the specimen was divided into three layers, where each layer was compacted 20 times, with scratching between each layer. Finally, the prepared specimen underwent vacuum saturation to keep the saturation of the CGM at 85%.

### 2.4. Experimental Scheme

According to engineering and research requirements, gravel content was set as either 0%, 10%, 20%, 30%, 40%, 50%, or 60%, which was the mass ratio of gravels to clay; the gravel shape was divided into either angular or round shape; the confining pressure was selected as 100 kPa, 200 kPa, or 300 kPa, recognized as pressure under 5 m, 10 m, or 15 m of ground, as shown in Figure 1b. The specific test scheme is shown in Table 2.

**Table 2.** Test scheme.

| Sample Number | Gravel Shape | Gravel Content | Confining Pressure σ (kPa) | Sample Number | Gravel Shape | Gravel Content | Confining Pressure σ (kPa) |
|---|---|---|---|---|---|---|---|
| Rc-0-0 | | 0% | 100, 200, 300 | Rc-1-0 | | 0% | 100, 200, 300 |
| Rc-0-1 | | 10% | 100, 200, 300 | Rc-1-0 | | 10% | 100, 200, 300 |
| Rc-0-2 | | 20% | 100, 200, 300 | Rc-1-0 | | 20% | 100, 200, 300 |
| Rc-0-3 | Round | 30% | 100, 200, 300 | Rc-1-0 | Angular | 30% | 100, 200, 300 |
| Rc-0-4 | | 40% | 100, 200, 300 | Rc-1-0 | | 40% | 100, 200, 300 |
| Rc-0-5 | | 50% | 100, 200, 300 | Rc-1-0 | | 50% | 100, 200, 300 |
| Rc-0-6 | | 60% | 100, 200, 300 | Rc-1-0 | | 60% | 100, 200, 300 |

## 3. Improved Resonant Column Device and Measuring Process

### 3.1. Description of the Improved Resonant Column

The resonant column device, which has a higher measuring accuracy when the dynamic strain ranges are within $10^{-6}$-$10^{-4}$, was made by the Nanjing University of Technology, Nanjing, China. As shown in Figure 4, the instrument consists of the following main components: (1) a confining pressure changing device, (2) a signal control device, (3) a driving device, (4) a draining condition control device, (5) an electrical signal enlarging device, and (6) a data collection device.

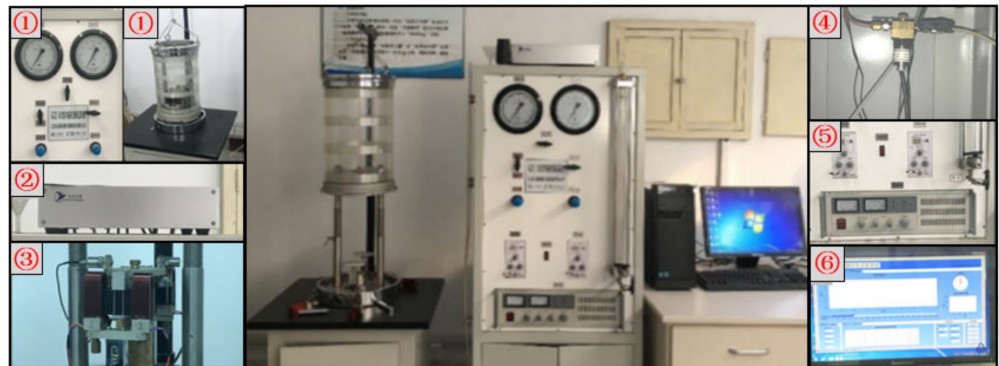

**Figure 4.** Resonant column device of GZZ-50 type [27,36].

For measuring the dynamic properties of a soft specimen that cannot stand without constraint [34], a hoop with a plug was added, as shown in Figure 5. This improvement enables the specimen to remain constrained (using either a split mold or confining pressure constraint) always, which better than freezes the specimen.

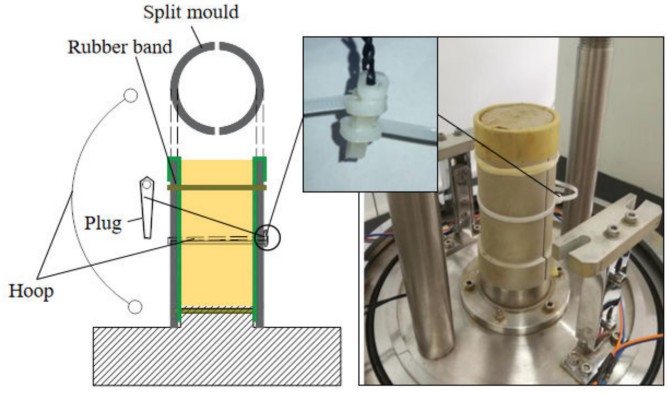

**Figure 5.** Schematic diagram of the process [27].

### 3.2. Testing Process

#### 3.2.1. Installation Process of the Specimen

The installation process of the specimen included five steps. Step one: measure the weight and height of the specimen, and put filter paper at the double end of the soil column. Step two: put the specimen at the base of the resonant column device and install the split mold and hoop with the plug. Step three: install the drive board at the top of the soil column and set the line to the designer line ring shown in Figure 6a. Step four: install the pressure chamber and set the confining pressure to solidify the specimen. Step five: rotate the control bar to pull out the plug and open the split mold, as shown in Figure 6b.

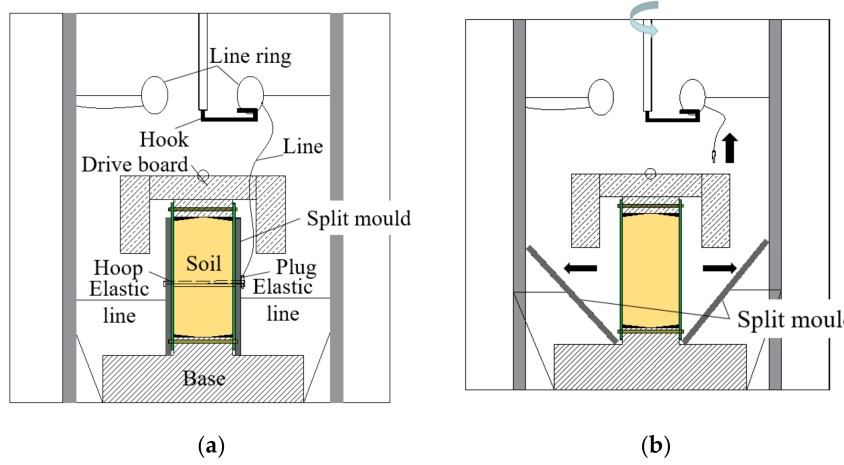

(**a**)  (**b**)

**Figure 6.** Schematic diagram of the installation process of the specimen.(**a**) Schematic diagram of the process in step three; (**b**) Schematic diagram of the process in step five [27].

#### 3.2.2. Testing Process and Data Deal

The functional division of the resonant column device of GZZ-50 type is shown in Figure 7.

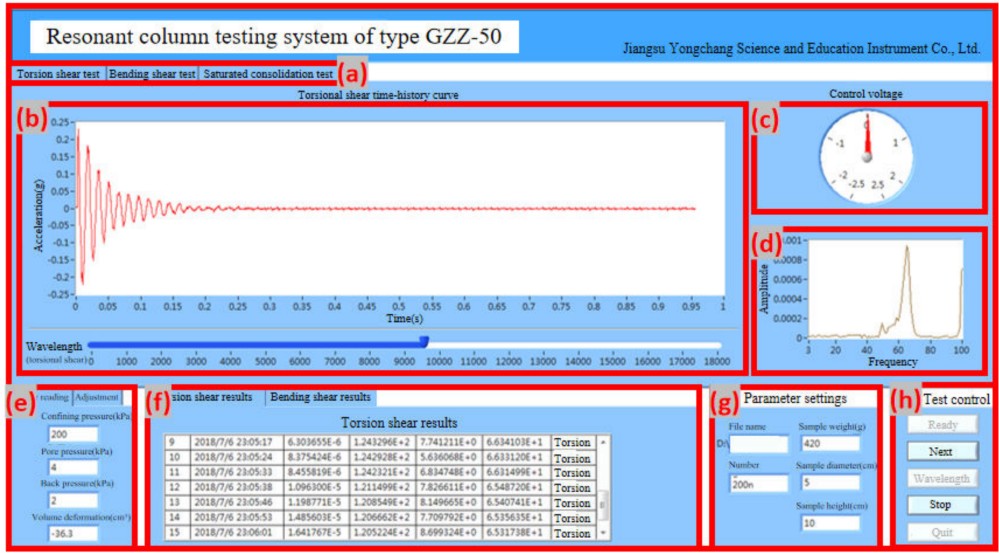

**Figure 7.** Functional division of the resonant column of GZZ-50 type. (**a**) choice of testing type; (**b**) data records of the accelerometer; (**c**) a set of exciting forces of the drive board; (**d**) Results of Fourier transform; (**e**) Statement information of the specimen; (**f**) Test results; (**g**) Data information; (**h**) Test controller.

Step one: Enter some basic physical information of the specimen, including diameter, height, and weight. Step two: Set the testing environment, including the confining pressure, drainage condition, and consolidation time. Step three: open the split mold by rotating the control bar to pull out the plug, as shown in Figure 5. Step four: Start the shear vibration and record the acceleration. Step five: deal with the obtained data. The dynamic shear modulus and the damping ratio were calculated according to Equations (1) and (2), respectively.

$$G = \rho \left( \frac{2\pi f h}{\beta} \right)^2 \tag{1}$$

In Equation (1), G is the dynamic shear modulus, MPa; $\rho$ is the density, g/cm$^3$; $f$ is the frequency of torsional vibration, Hz; h is the height of specimen, mm; and $\beta$ is the eigenvalue of the frequency equation of torsional vibration.

$$D = \frac{1}{2\pi} \frac{1}{m} In \frac{A_n}{A_{n+m}} \tag{2}$$

In Equation (2), m is the number of cycles between peaks $A_n$ and $A_{n+m}$; $A_n$ is the n-th amplitude of the peak after excitation; and $A_{n+m}$ is the m-th amplitude of the peak after the n-th amplitude.

## 4. Results and Discussion about the Dynamic Shear Modulus

### 4.1. Dynamic Shear Modulus

For investigating the effects of confining pressure, gravel content, and shape on the dynamic shear modulus of CGMs, the relationships of the dynamic shear modulus (G) with shear strain (γ) are provided in Figure 8. In this study, all of the results of the CGMs are available, and only the representative test results were selected and analyzed.

Figure 8 shows the relationships of the dynamic shear modulus with shear strain for different confining pressures (100, 200, and 300 kPa) and gravel shapes (round and angular). For the given gravel content and shape, it can be observed that the dynamic shear modulus increased with the increase of confining pressure. The reason for the increase in the dynamic shear modulus caused by the confining pressure is due to the further compaction of the specimen, which will lead to (1) a decrease in the void ratio, (2) an increase in the relative density, and (3) an improvement in the contacting points of the particles [8].

As shown in Figure 8, comparing the dynamic shear modulus with different gravel contents of the same gravel shape and confining pressure, a lower increase in amplitude appears between CGMs with 0% and 10% gravel content, and a large increase in amplitude appears when the gravel content is over 10% in CGMs. It can be explained that, with the increase of gravel content, improvements in the CGM will gradually appear due to the better shear wave transfer effect of gravels than of clay when the gravel content is over 10%, and the specific reason is as follows.

Figures 9 and 10 display the results of the Microscope diagram of clay and the Computer Tomography (CT) diagram of CGM, respectively. It can be concluded from Figure 9 that the clay consists of lots of particles, and that the size of particles is near 4 µm. According to the CT diagram of the CGM in Figure 10, it can be found that, with the increase of gravel content, more gravels appear in the section diagram of the CGM, and that the section size of the gravels is in the range of 2 and 10 mm. In addition, the angular gravels tend to have less relative distance between each other, which can be recognized as an interlock effect [2,17]. The large size of the gravels reduces the amount of clay particles and the contacting points of particles in the given volume, and the high stiffness of the gravels increases the propagation effect of the stress wave. Hence, with the increase of gravel content, the dynamic shear modulus increases in total tendency.

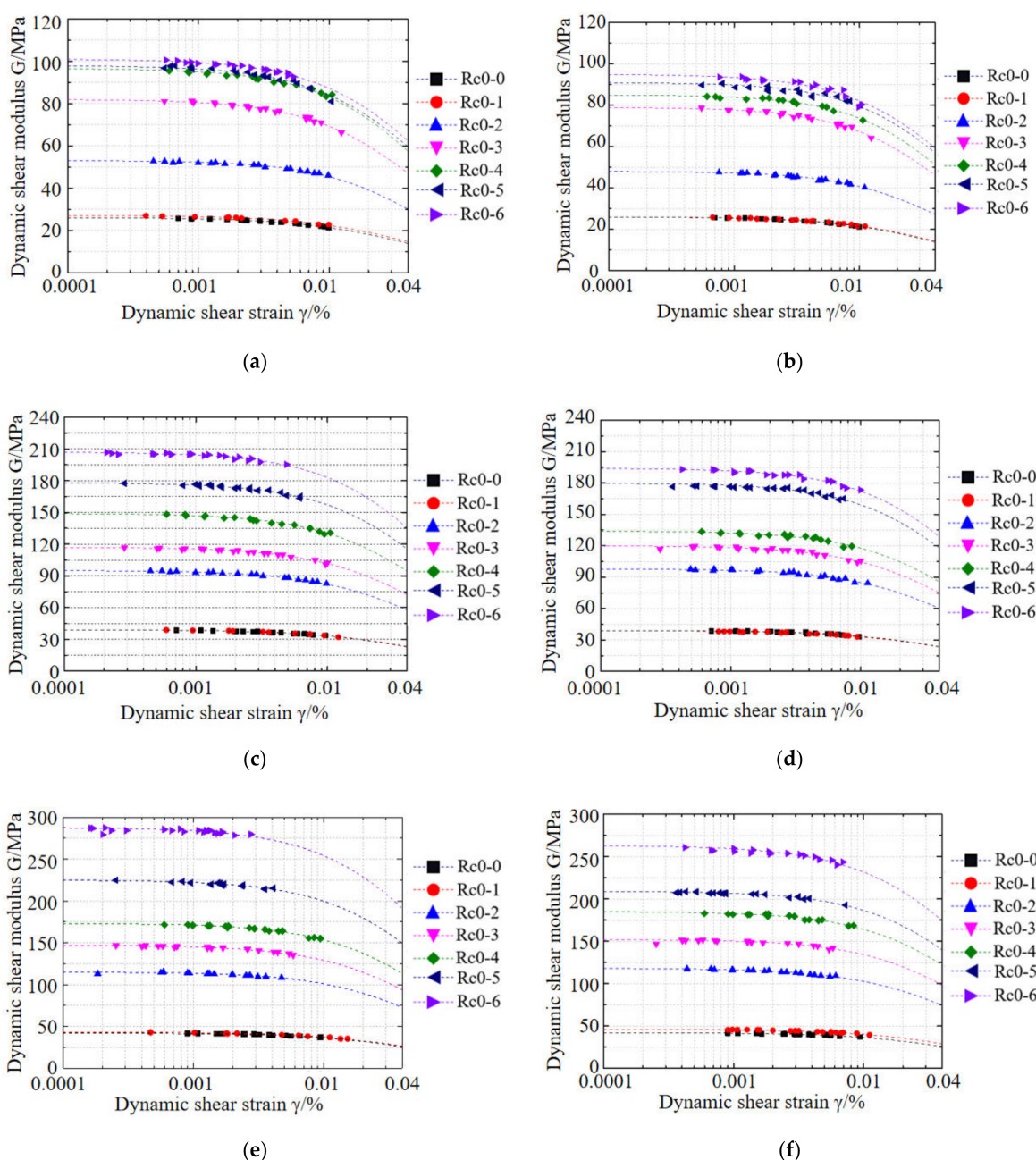

**Figure 8.** The variation of the dynamic shear modulus with dynamic shear strain; (**a**) clay–gravel mixture (CGM) (Rc-0-100 kPa); (**b**) CGM (Rc-1-100 kPa); (**c**) CGM (Rc-0-200 kPa); (**d**) CGM (Rc-1-200 kPa); (**e**) CGM (Rc-0-300 kPa); (**f**) CGM (Rc-1-300 kPa).

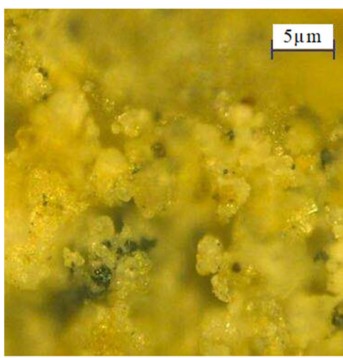

**Figure 9.** Microscope diagram of the clay.

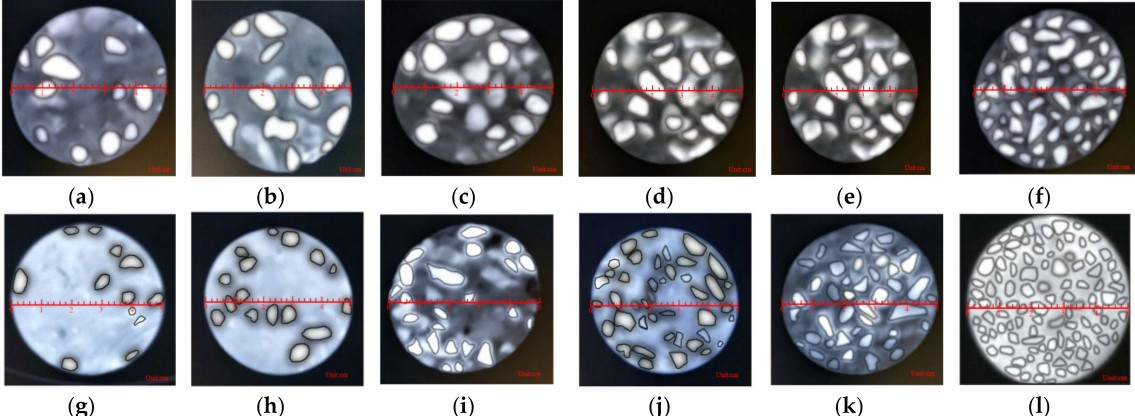

**Figure 10.** Computer Tomography diagram of the CGM. (**a**) Rc-0-1; (**b**) Rc-0-2; (**c**) Rc-0-3; (**d**) Rc-0-4; (**e**) Rc-0-5; (**f**) Rc-0-6; (**g**) Rc-1-1; (**h**) Rc-1-2; (**i**) Rc-1-3; (**j**) Rc-1-4; (**k**) Rc-1-5; (**l**) Rc-1-6.

As shown in Figure 8, comparing the dynamic shear modulus of CGMa and CGMr, it can be observed that CGMr has a larger dynamic shear modulus than CGMa when the dynamic strain is smaller than $10^{-5}$. Furthermore, with the increase of shear strain, it can also be found that the reducing amplitude of CGMa is smaller than of CGMr under the given confining pressure.

The reason for the greater dynamic shear modulus of CGMr under lower shear strain ($<10^{-5}$) is because of the better compaction characteristic of CGMr under the same compaction work (shown in Figure 11) and the better completion in the gravel's edges (shown in Figure 3) [16,17,20,35]. The lower reduction tendency of the dynamic shear modulus with shear strain in CGMa can be explained by the better interlock effect between angular gravels when the dynamic shear strain appears in the CGM [2,17].

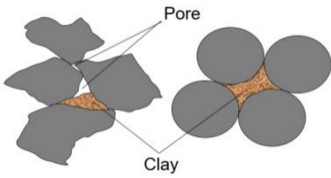

**Figure 11.** Schematic diagram of the compaction results in the Clay–gravel mixtures (CGMs.).

*4.2. Maximum Dynamic Shear Modulus*

The maximum dynamic shear modulus ($G_{max}$) of the CGMs, at a strain level of $10^{-6}$, under three different confining pressures (100, 200, and 300 kPa), seven different gravel contents (0%, 10%, 20%,

30%, 40%, 50%, and 60%), and two different gravel shapes (round and angular shape), were studied. Hardin and Drnevich [36,37] proposed a fitting equation (Equation (3)) according to the hyperbolic stress–strain relationships, which can compute the dynamic shear modulus (G) at any strain (γ).

$$\frac{G}{G_{max}} = \frac{1}{1 + \left(\frac{\gamma}{\gamma_r}\right)^a} \tag{3}$$

In Equation (3), $G_{max}$ is a curve-fitting parameter, named the maximum dynamic shear modulus, MPa; $\gamma_r$ is a curve-fitting parameter, named the referent shear strain; and a is a curve-fitting parameter, named curvature parameter. Based on the different data points, the value of this parameter is different, but it is within the range of 0.998 ± 0.003. In order to compare the maximum dynamic shear modulus and the reference shear strain, the value of this parameter was taken uniformly as 0.998 in the following discussion.

Figure 12 shows the variation of $G_{max}$ with different gravel contents for different confining pressures and gravel shapes. It can be observed that, in total, the $G_{max}$ of the CGM increased with the increase of gravel content, but the increasing amplitude was different. When the gravel content was limited between 0% and 10%, a lower increase in amplitude appeared; when between 10% and 40%, a greater increase in amplitude appeared; and when the gravel content was over 40%, different increases in amplitude under different confining pressures were observed, as well as less amplitude under 100 kPa, medium amplitude under 200 kPa, and greater amplitude under 300 kPa.

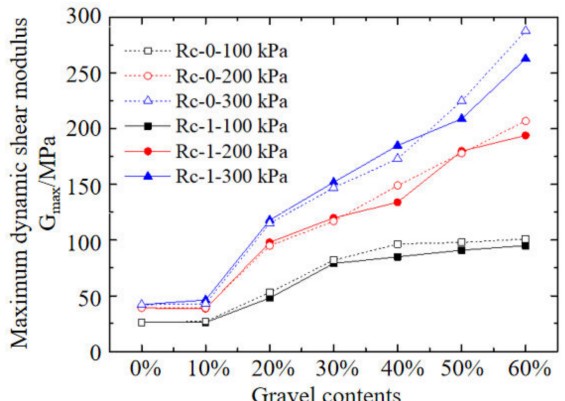

**Figure 12.** The variation of the maximum dynamic shear modulus with different gravel contents.

In addition, when the gravel content was between 50% and 60%, it can also be found that the $G_{max}$ of CGMr was greater than CGMa, especially under 300 kPa confining pressure. Similar results were obtained by Meidani et al. [16,17], and the reason for this is that, with high gravel content (more than 50%), the gravels make contact with each other, and the shape advantages of round gravels appear in (1) better edge completion, (2) better clay particle filling effects under the same compaction work, and (3) better transfer effects for shear stress wave [8,37,38].

For predicting $G_{max}$ in CGMs with different gravel contents, gravel shapes, and confining pressures, a predicting equation was put forward, as shown in Equation (4).

$$G_{max} = 110 + 56x + 33c + (1.4 - 2.8s)x^2 + 22xc$$
$$R^2 = 0.96 \tag{4}$$

In Equation (4), x is in the range of 0 to 0.6, named the gravel content; c is in the range of 0 to 3, named the normalized confining pressure (the confining pressure/100 kPa); and s is in the range of 0 to 1, named the shape characteristic of gravels, in which 0 represents CGMr and 1 represents CGMa.

### 4.3. Referent Shear Strain

According to Equation (3), the relationship of the referent shear strain $\gamma_r$ (obtained from curve fitting) with gravel contents for different confining pressures and gravel shapes is shown in Figure 13. Based on Equation (3), it can be concluded that the greater the referent shear strain, the lower the reduction amplitude of the dynamic shear modulus with shear strain.

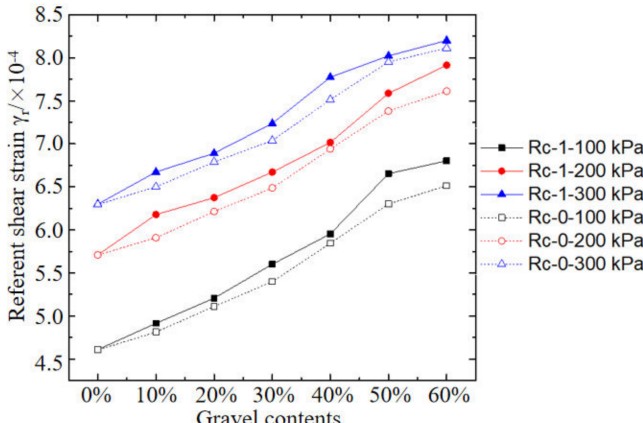

**Figure 13.** The variation of the referent shear strain with different gravel contents.

Figure 13 shows that the referent shear strain increases with the increase of gravel content at the given confining pressures and with different gravel shapes. A similar tendency was observed by Meidani et al. [11,16,17], and it can be explained by the large size and the high strength of the gravels. Thelarger the size of the gravels, the less contacting points in the CGM at the given volume, and the higher the strength, the better the gravels can transfer the shear wave than clay [39,40]. Hence, a greater referent shear strain appeared in the CGMs with the increase of gravel contents.

Comparing the referent shear strain with the different gravel shapes with the given gravel content and confining pressure, it can be observed that the referent shear strain of CGMa is greater than CGMr under the same confining pressure and with the same gravel content. The reason for this tendency is mainly focused on the interlock effect between the particles [2,17], in which angular gravels limit particles to dislocate or rotate relatively, whereas round gravels enhance this ability in CGM. Therefore, a greater referent shear strain appeared in CGMa.

Furthermore, it can also be found in Figure 13 that the referent shear strain increases with the increase of confining pressure with the given gravel contents and shapes. A decrease in the void ratio, an increase in the relative density, and an increase in the contacting points of the particle can be used to explain this tendency well.

For predicting $\gamma_r$ in CGMs with different gravel contents, gravel shapes, and confining pressures, a predicting equation was put forward, as shown in Equation (5).

$$\gamma_r = (3.8 + (3.3 + 0.3s)x + 0.82c) \times 10^{-4}$$
$$R^2 = 0.97 \tag{5}$$

In Equation (5), x is in the range of 0 to 0.6, named the gravel content; c is in the range of 0 to 3, named the normalized confining pressure (the confining pressure/100 kPa); and s is in the range of 0 to 1, named the shape characteristic of gravels, in which 0 represents CGMr and 1 represents CGMa.

## 5. Results and Discussion about the Damping Ratio

### 5.1. Damping Ratio

The damping ratio (D) is an important parameter in characterizing the energy dissipation characteristics of a material. For all of the tested CGM specimens, the relationships of the damping ratio with shear strain for different values of confining pressure, gravel content, and gravel shape were studied. In order to explore the effects of confining pressure, gravel content, and shape on the damping ratio, only some representative test results were selected, but the others also showed similar trends.

Figure 14 shows that the damping ratio of the CGMs increases nonlinearly with the increase of the shear strain. When the CGMs are under small shear strain ($<10^{-5}$), the damping ratio increases gently, and then the damping ratio sharply increases. A similar tendency was verified in sand by Shi et al. [41–43]. In CGMs, when dynamic shear strain appears, the particles in the mixtures relatively dislocate or rotate, which consumes the stress wave energy and leads to damping. With an increase of shear strain, a more relative dislocation or rotation appears between the particles in the CGMs, and a larger damping shows.

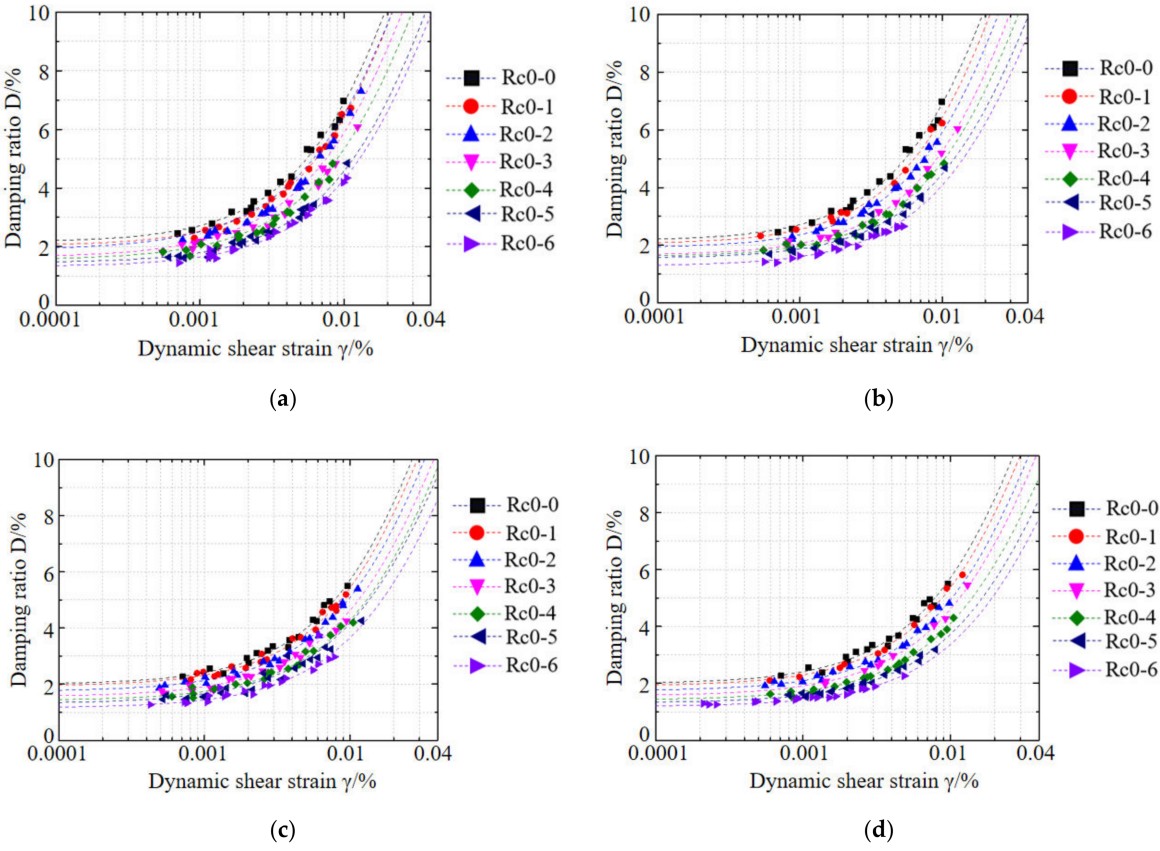

**Figure 14.** *Cont.*

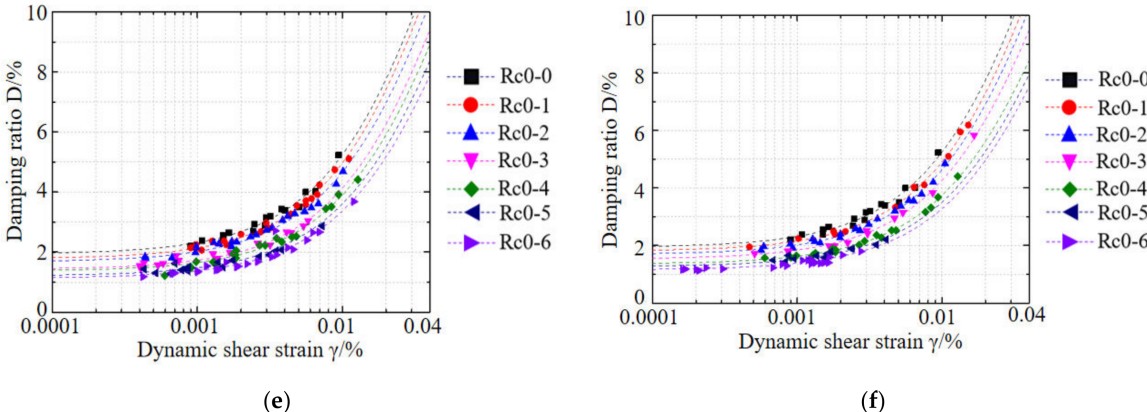

(**e**)
(**f**)

**Figure 14.** The variation of the damping ratio with dynamic shear strain. (**a**) CGM (Rc-0-100 kPa); (**b**) CGM (Rc-1-100 kPa); (**c**) CGM (Rc-0-200 kPa); (**d**) CGM (Rc-1-200 kPa); (**e**) CGM (Rc-0-300 kPa); (**f**) CGM (Rc-1-300 kPa).

Figure 14 also depicts the variation of the damping ratio with confining pressure for different gravel contents and shapes. It is observed that with an increase in confining pressure, the damping ratio of CGMs decreases with the given gravel contents and shapes, and the reduction amplitude gradually decreases too. A similar tendency of the confining pressure to damping ratio in CGMs/sand rubber mixtures/gravel rubber mixtures was obtained by Meidani et al. [7,16,17,31], and the reason for this is due to an increase in the relative density of the CGMs that limits the relative dislocation or rotation of particles, which reduces the energy consumption (meaning a lower damping ratio). However, with an increase in the relative density of CGMs, a greater confining pressure is needed to increase the relative density during consolidation; hence, the reduction tendency of the damping ratio shows nonlinearity.

Comparing the damping ratio with the different gravel contents for the given gravel shape and confining pressure, it was observed that the damping ratio of the CGMs reduced with the increase of gravel content. This tendency was similar to the results in sand with different fine contents obtained by Wang et al. [11,16]. With an increase in gravel content, a greater volume of the CGMs in the given space is occupied by the gravels, which means less contacting points between the particles due to the large size of gravels; hence, a lower damping ratio appeared with the increase of gravel content.

*5.2. Minimum Damping Ratio*

The variation of the minimum damping ratio ($D_{min}$) with different gravel contents for different confining pressures and gravel shapes was investigated in this study. Based on Hardins' predicting equation, the relation expression of the damping curve was modified by Chen et al. [44], and the expressions are shown as follows.

$$D = D_{min} + D_{max}\left(\frac{\gamma/\gamma_r}{1+\gamma/\gamma_r}\right)^m \tag{6}$$

In Equation (6), $D_{min}$ is a curve-fitting parameter, named the minimum damping ratio; $D_{max}$ is a curve-fitting parameter, named the maximum damping ratio; m is a curve-fitting, number named the named curvature parameter, which was 0.997 in this article; and $\gamma_r$ is an obtained parameter during the dynamic shear modulus analysis.

The comparison of the $D_{min}$ with gravel content for different confining pressures and gravel contents is provided in Figure 15. For the given confining pressure and gravel shape, the $D_{min}$ of the CGMs decreases with the increase of gravel content, and a similar tendency was observed by

Sun et al. [16]. The reason for the decrease of the $D_{min}$ with the increase of gravel content is due to a decrease in the amount of contacting points and the better shear stress wave transfer effect of gravels.

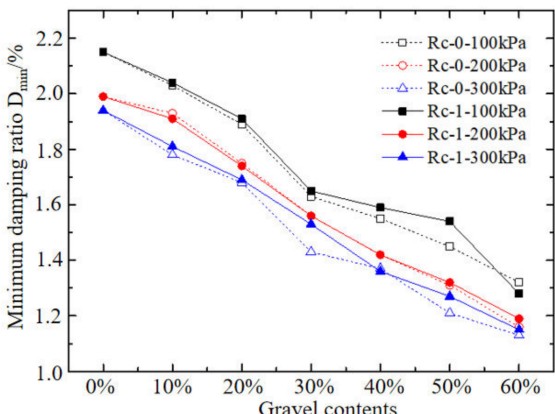

**Figure 15.** The variation of the minimum damping ratio with dynamic shear strain.

Figure 15 also shows the relationship of the $D_{min}$ and the CGMs with different gravel shapes for the given gravel content and confining pressure. It can be observed that, in total, the $D_{min}$ of CGMa is smaller than CGMr, owing to (1) better edge completion, (2) better clay particle filling effects under the same compaction work, and (3) better transfer effects for shear stress wave [8,38,41]. It can also be found that with the increase of confining pressure, the $D_{min}$ of the CGMs nonlinearly reduces, and a similar tendency was obtained in the CGMs and sand–gravel mixtures by Meidani et al. [17,44]. This tendency is due to (1) a decrease in the void ratio, (2) an increase in the relative density, and (3) an increase in the contacting points of the particles [8].

For predicting the $D_{min}$ in CGMs with different gravel contents, gravel shapes, and confining pressures, a predicting equation was put forward, as shown in Equation (7).

$$D_{min} = 2.24 - 1.4x - (1.1 - s)c$$
$$R^2 = 0.98 \tag{7}$$

In Equation (7), x is the gravel content (0–0.6); c is the normalized confining pressure (the confining pressure/100 kPa, 0–3); s is the shape characteristic of the gravels (0–1, in which 0 and 1 represent CGMr and CGMa, respectively).

*5.3. Maximum Damping Ratio*

Based on Equation (6), the relationship of the maximum damping ratio $D_{max}$ (results from curve fitting) and gravel contents for different confining pressures and gravel shapes were obtained as shown in Figure 16.

As shown in Figure 16, it can be concluded that the maximum damping ratio reduces with the increase of gravel content for the given gravel shape and confining pressure. This tendency is mainly due to the greater shear wave transfer effect of gravels than clay.

Comparing the results of the maximum damping ratio with the different gravel shapes in Figure 16, it can be observed that CGMr has a larger maximum damping ratio than CGMa when the gravel content is over 30%. Combined with Sun's results [16,17,35], the particles with angular shape tend to limit dislocation or rotation in CGMs, which relatively reduced the consumption of the shear wave energy and shows a smaller damping ratio under relatively large shear strain. In addition, with the increase of confining pressure, the maximum damping ratio reduces nonlinearly, where a similar tendency of the confining pressure to the maximum damping ratio of soil has been concluded by

Lin et al. [6,13,45], and the corresponding causes focus on the reduction of the void ratio and the increase of relative density.

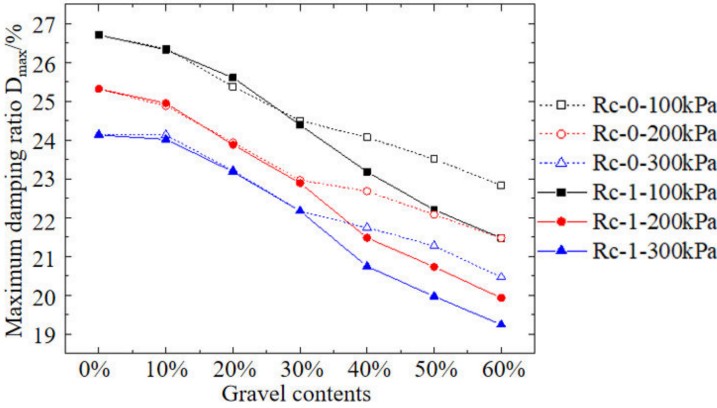

**Figure 16.** The variation of the maximum damping ratio with dynamic shear strain.

For predicting the $D_{max}$ in CGMs with different gravel contents, gravel shapes, and confining pressures, a predicting equation was put forward, as shown in Equation (8).

$$D_{max} = 28 - 7.6x - 1.2c + (1.2 - 4.4s)x^2$$
$$R^2 = 0.97$$

(8)

In Equation (8), x is the gravel content (0–0.6); c is the normalized confining pressure (the confining pressure/100 kPa, 0–3); and s is the shape characteristic of the gravels (0–1, where 0 and 1 represent CGMr and CGMa, respectively).

## 6. Conclusions

Based on subgrade filling engineering with CGMs in Ganzhou, Jiangxi province, China, 42 groups of resonant column tests were performed to investigate the low-strain dynamic characteristic of CGMs under the same compacting power, and the following conclusions could be drawn from this study.

(1) For measuring the dynamic properties of soft materials (those that cannot stand without confining pressure) at low strain, improvements regarding the resonant column of GZZ-50 type were conducted and the corresponding installation and data deal processes were detailed.

(2) The relationship of the dynamic shear modulus and the damping ratio with different confining pressures and gravel contents were obtained, in which, with the increase of confining pressure and gravel content in the CGMs, the dynamic shear modulus shows an increasing tendency and the damping ratio shows a decreasing tendency to different degrees.

(3) According to Hardin's fitting results, the relationship of $G_{max}$ and $\gamma_r$ with different confining pressures, gravel contents, and shapes was obtained. With the increase of gravel content, $G_{max}$ shows three different increasing phases, including 0–10% (low increasing amplitude), 10–40% (high increasing amplitude), and 40–60% (increasing amplitude seriously affected by confining pressure), owing to the better shear wave transfer effect of gravels. With the increase of confining pressure, $G_{max}$ shows a nonlinear increasing tendency that is related to the compaction effect of CGMs. CGMr has greater $G_{max}$ than CGMa when the gravel content is over 50%, and less of a difference appears when the gravel content is smaller than 50%.

(4) The $\gamma_r$ of the CGMs linearly increases with an increase in gravel content, and shows a nonlinear increase with the increase of confining pressure; in addition, the $\gamma_r$ of CGMa is greater than CGMr, which is due to the interlock effect of angular gravels.

(5)   Based on the testing results and Chen's fitting equation, the $D_{min}$ of the CGMs linearly decreases with the increase in gravel contents, and nonlinearly decreases with the increase in confining pressure; CGMr has a lower $D_{min}$ than CGMa.

(6)   Based on Chen's fitting equation, the $D_{max}$ of the CGMs decreases with an increase in gravel content and confining pressure; CGMa has a smaller $D_{max}$ than CGMr when the gravel content is over 30%, and the influence of gravel shape on the $D_{max}$ of the CGMs is not clear when the gravel content is smaller than 30%.

(7)   For better supporting the design of subgrade filling engineering with CGMs, the prediction equations of $G_{max}$, $\gamma_r$, $D_{min}$, and $D_{max}$, considering different confining pressures (0–300kPa), gravel contents (0–60%), and shapes (round and angular), were obtained.

It should be noted that in this paper, the dynamic characteristics of saturated soil–rock mixture under low strain were studied only by resonant column. In order to better serve the project, the water content in the soil–rock mixture should be taken into account to explore its dynamic characteristics. What is more, dynamic triaxial and local strain tests should be conducted to understand the dynamic characteristics with different strain ranges.

**Author Contributions:** All authors contributed substantially to this study. Individual contributions: investigation, X.H.; writing—original draft, A.Z.; writing—review and editing, W.W.; validation, P.J. All authors have read and agree to the published version of the manuscript.

**Funding:** This research was funded by the National Natural Science Foundation of China (Grant numbers 51579119 and 41772311), the Natural Science Fund for Colleges Universities of Jiangsu Province (Grant number 17KJB560003), the Zhejiang Provincial Natural Science Foundation of China (Grant number LY17E080016), and the Research Fund of the Zhejiang Provincial Department of Housing and Urban–Rural Development (grant numbers 2017K179 and 2016K130).

**Conflicts of Interest:** The authors declare no conflicts of interest.

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
