# Peer review of "Characterization of the Dynamic Properties of Clay–Gravel Mixtures at Low Strain Level"

_sustainability, doi:10.3390/su12041616_

Round 1
Reviewer 1 Report
This work presented in this study is interesting and well presented. The analysis of the results is satisfactory.
However, English must imperatively be revised in depth since there are many grammatical and spelling mistakes. In addition, there is a lot of typographical error (badly positioned dots or commas, unnecessary capital letters, etc.). Some mistakes are cited below, but not all of them because they are too many. It is therefore necessary to make a correction and a careful verification of the text.
Abstract : The numbering of testing results is not appropriate.
Introduction : The scientific contribution of your work compared to previous studies is not clearly explained in the introduction. Please give details on this point.
Conclusion : Please give some outlooks on your work after the conclusions.
Table 1 : Please give the standards used for the soil characterization. Please give the standard deviation of every properties.
Figure 5 : Picture should be bigger because it is too small. You must keep the original aspect ratio of pictures, or crop them, but do not distort the images. Add a dot at the end of the sentence of the figure title.
Figure 7 : Move the reference citation [27] at the end of the sentence title, just before the dot point.
Figure 9 and 14 : It would be better to enlarge the font size of the X and Y axes legends of each graph (a to f). In fig. 9, in the Y-axis : correct MPa and not Mpa.
Figure 10 : Please delete the dot on the right side of the picture. Add a dot at the end of the sentence of the figure title.
Figure 14 : There is 2 figures 14. Please correct it.
Figures 13, 14 (the 1st) and 15 : You should homogenize font size of the X and axes legends of each graph between every figures.
Line 188 : delete the dot between “Figures” and “10” and put a space.
Line 208 : Add a space between “200” and “and”.
Line 214 : Where does the curvature parameter come from ? Why is it 0,998 for this study ?
Line 228 : Move the references in the phase, do not put them after the dot.
Line 302 : Please correct the word “difffernt”. Delete the capital letter “Comparison”.
Reviewer 2 Report
The authors present a paper on the characterization of dynamic properties of clay-gravel mixtures at low strain level. The topic is interesting and the paper in general is well structured and well organized. The work plan is correct, and the results analysis is also well structured. In my opinion, the paper has conditions to be considered for publication. I only consider that the following information needs to be included in the abstract: should mention motivation and expected impact of the research.
